# Analysis of Near-Infrared Spectral Properties and Quantitative Detection of *Rose Oxide* in Wine

Xuebing Bai [1,2], Yaqiang Xu [1], Xinlong Chen [1], Binxiu Dai [1], Yongsheng Tao [1,2,*] and Xiaolin Xiong [3,*]

[1] College of Enology, Northwest A&F University, Xianyang 712100, China; bxb@nwafu.edu.cn (X.B.); xuyaqiang@nwafu.edu.cn (Y.X.); daibx@nwafu.edu.cn (B.D.)
[2] Ningxia Helan Mountain's East Foothill Wine Experiment and Demonstration Station of Northwest A&F University, Yongning 750104, China
[3] Xue Lin Yuan (Shenzhen) Wine Culture Co., Ltd., Shenzhen 518000, China
* Correspondence: taoyongsheng@nwsuaf.edu.cn (Y.T.); somso99999@gmail.com (X.X.)

**Abstract:** This study aims to investigate the near-infrared spectral properties of *Rose Oxide* (4-Methyl-2-(2-methyl-1-propenyl) tetrahydropyran) in wine, establish a quantitative detection, and build relationships between the chemical groups of *Rose Oxide* and near-infrared characteristic bands, so as to provide ideas and references for the near-infrared detection of a low-content aroma substance in wine. In total, 133 samples with different wine matrices were analyzed using Fourier transform–near-infrared (FT-NIR) spectroscopy. Min–max normalization (MMN), principal component analysis (PCA), and synergy interval partial least squares regression (Si-PLSR) were used for pre-processing, outlier rejection, analysis of spectral properties, and modeling. Finally, the quantitative detection model was established using the PLSR method and the wine sample containing *Rose Oxide* was verified externally. Eight subintervals (4000–4400 cm$^{-1}$, 4400–4800 cm$^{-1}$, 5600–6000 cm$^{-1}$, 6000–6400 cm$^{-1}$, 6400–6800 cm$^{-1}$, 6800–7200 cm$^{-1}$, 7200–7600 cm$^{-1}$, 8400–8800 cm$^{-1}$) were determined as the characteristic band intervals of *Rose Oxide* in the NIR region. Among them, 5600–6000 cm$^{-1}$ was assigned to the first overtone C–H stretching in tetrahydropyran ring and methyl as well as the combination C–H stretching of the CH$_3$ function groups, 6000–6400 cm$^{-1}$ was assigned to the first overtone C–H stretching of the C–H=group and the combination C=C stretching in isobutyl, and 8400–8800 cm$^{-1}$ was assigned to the second overtone C–H stretching and C–O stretching in tetrahydropyran ring as well as the C–H stretching vibration in methyl. In addition, 4000–4800 cm$^{-1}$, 6400–6800 cm$^{-1}$, and 7200–7600 cm$^{-1}$ were assigned to the C–H stretching vibration, while 6400–7600 cm$^{-1}$ was assigned to the C–O stretching vibration. The training result showed that the calibration model ($r_{cv}^2$ of 0.96 and $RMSE_{CV}$ of 2.33) and external validation model ($r_{cv}^2$ of 0.84 and $RMSE_{CV}$ of 2.72) of *Rose Oxide* in wine were acceptable, indicating a good predictive ability. The spectral assignment of *Rose Oxide* provides a new way for the NIR study of other terpenes in wine, and the use of the established Si-PLSR model for the rapid determination of *Rose Oxide* content in wine is feasible.

**Keywords:** *Rose Oxide* (4-Methyl-2-(2-methyl-1-propenyl) tetrahydropyran); de-aromatic wine; NIR spectroscopy; Si-PLSR; wavebands analysis



## 1. Introduction

*Rose Oxide* (4-Methyl-2-(2-methyl-1-propenyl) tetrahydropyran), with strong fragrance of rose and lychee, is the main component of rose and rose geranium. It is not only used to prepare flavors, such as rose, leaf, and other flower flavors, but it is also widely used in upscale cosmetics and the food industry. Surprisingly, many varieties of grapes also contain *Rose Oxide* broadly, which enriches their aroma features and makes wine purer and fresher [1]. As an oxide of monoterpenols, *Rose Oxide* has a strong volatility. It becomes a kind of recognizable aroma [2]; even its concentration reduced by three times during fermentation [3]. Therefore, *Rose Oxide* is an important sign for identifying the varieties, years, and origins of wine objectively [4]. Studies have shown that the existence of *Rose*

*Oxide* is highly significant correlated with whether grape and wine have a rose aroma [1]. Although *Rose Oxide* has a low concentration in wine, it also has a lower odor threshold with only 0.2 μg/L [4] and a higher odor activity value (OAV) of generally more than 100 [4,5]. This means that *Rose Oxide* could be easily perceived and contribute significantly to the overall aroma of wine. In addition, *Rose Oxide* could react with other terpene aroma substances and play a decisive role in the formation of aroma when other aroma substances hold a low concentration [2]. Many studies have revealed that *Rose Oxide* is the key compound for bringing out the floral, rose, and even lychee aromas of wine [6,7] and that it correlates with positive emotions and higher liking scores for wine consumers [8].

However, the quantification of *Rose Oxide* is complicated as a trace component. Gas chromatography (GC) coupled to at least one detector, such as a flame ionization detector (GC-FID) or a mass spectrometer (GC-MS), is the typical method for analyzing the *Rose Oxide* content in wine [9]. It is necessary to conduct a pre-treatment of the wine sample, such as extraction, heating, oscillation, analysis, as well as other steps, and finally use the instrument to conduct a qualitative and quantitative analysis. These methods are labor-intensive and costly, and they easily cause the loss of volatility of *Rose Oxide*. Therefore, a rapid, simple, and economical method for predicting the content of *Rose Oxide* as an alternative to the traditional analysis methods is required. Near-infrared (NIR) spectroscopy can address these limitations.

The NIR spectrum lies between the visible and IR regions of the electromagnetic spectrum in the wavelength range 780–2500 nm and involves the excitation of non-fundamental vibrations, overtones, and combination modes [10,11]. NIR spectroscopy mainly reflects the information of hydrogen-containing groups, including C–H (such as methyl, methylene, methoxy, carboxyl, and so on), O–H (hydroxyl), S–H (sulfhydryl), N–H (amino), and so on. There is also some other groups' information (such as C=C, C=O, and so on), but the intensity is weak. These groups are important components of organic compounds, which means that the structures and compositions of almost all organic compounds can be found in the near-infrared spectrum. The process of NIR generally includes spectrum pretreatment, outlier elimination, band screening, and quantitative model establishment. Band screening, also called spectrum allocation, aims to detect the feature wavebands of the chemical groups in targeting ingredients and ensuring the spectral fingerprint information of the ingredients. It is the basis for establishing the quantitative models and providing models with a theoretical note. NIR spectroscopy is a simple and non-destructive technique which generally does not require any sample pretreatment which may result in the loss of the substance under test. Therefore, it is widely used in agriculture, petroleum, chemical, tobacco, pharmaceutical, and food industries [12]. Several studies have used NIR spectroscopy for predicting compounds in wine, such as phenolic compounds [13], trace metal elements [14], and volatile compounds [9], as well as different terpenes in plants, such as α-pinene, β-pinene, myrcene, eugenol, cineole, and linalool [11,15]. To the best of our knowledge, no attempts have been made to establish an association between the NIR spectrum and terpene profiles of wines, let alone study the spectral characteristics of its molecular group and create a quick detection method in wines.

Here, we aimed to investigate the spectral properties of *Rose Oxide* and methods of quantitatively detecting it in wine. First, based on a single controlled environment (model wine), the NIR feature wavebands of *Rose Oxide* were screened using spectral preprocessing, outlier rejection, and synergy interval partial least squares (Si-PLS) methods. Second, a quantitative detection method for *Rose Oxide* was constructed based on these spectral wavebands using the partial least squares regression (PLSR) method in a relatively complex environment (de-aromatic wine). Third, wine samples containing *Rose Oxide* were used to effectively validate the accuracy and model transferability of the above method. In this study, NIR spectroscopy was used to analyze the typical terpene compounds in wine, establish a rapid detection method of *Rose Oxide*, and hopefully provide methodological support for the rapid and non-destructive detection of terpene compounds in wine.

## 2. Materials and Methods

### 2.1. Materials

Grape variety to be de-flavored: Cabernet Franc, collected from the Ningxia Helan Mountain's East Foothill Wine Experiment and Demonstration Station of Northwest A&F University in October 2021, which contains 222.5 g/L of residual sugar (expressed as glucose) and 4.6 g/L of acid (expressed as tartaric acid); bacterial strain: a strain of *S. cerevisiae* called ACTIFLORE F33 from Lafford Company in France; and the sample set of external verification: 21 Cabernet Franc dry red wines (produced from wineries at Ningxia Helan Mountain's East Foothill) with different concentrations of *Rose Oxide* were added to construct external verification wine samples.

### 2.2. Instruments and Reagents

The instruments and reagents used in the experiment include: HW.SY21-KP8 Electric Thermostatic Water Bath (Chengfeng Inc., Beijing, China ); ME203E Electronic Balance (Mettler Toledo Inc., Shanghai, China); FE28pH meter (Mettler Toledo Inc., Shanghai, China); DW-YL270 Cryogenic Refrigerator (Zhongke Meiling Cryogenic Technology Inc., Hefei, Anhui, China); KH-500DE CNC ultrasonic cleaner (Hechuang Ultrasonic Instrument Inc., Kunshan, Jiangsu, China); Hei-VAP Table Rotary Evaporation Instrument (Hadolf Instrument Equipment Inc., Shanghai, China); GCMS-QP2020 Gas chromatography–mass spectrometry instrument (Shimazu Laboratory Equipment Inc., Shanghai, China); Bruker-TANGO-T Fourier Transform–near-infrared spectrometer (Brock Scientific Instruments Inc., Hong Kong, China), a built-in automatic background scanning program can timely eliminate the impact of environmental changes in detection results, equipped with Rock-SolidTM patent interferometer, multi-layer coating low OH quartz beam splitter, and InGAs digital detector.

The ultrapure water was obtained from the Milli-Q Pure Water Preparation System (Millipore Inc., Molsheim, France). The analytically pure-grade reagent, anhydrous ethanol, tartaric acid, and sodium hydroxide were purchased from Chemical Reagent Inc., Tianjin, China. The chromatographic grade reagents, 2-octanol (purity ≥ 99.0%) and (+)-*Rose Oxide* (purity ≥ 99.0%), were purchased from Sigma-Aldrich Corporation (Beijing, China).

### 2.3. Methods

#### 2.3.1. Sample Preparation and Data Acquisition

Model wine preparation: In total, 120 mL of anhydrous ethanol, 880 mL of distilled water, and 5 g of tartaric acid were added to the blue silk-mouthed bottle and mixed evenly with ultrasonic waves; the pH was adjusted to between 3.2 and 3.4 using saturated NaOH. This was followed by the addition of *cis-Rose Oxide* into the configured model wine to the concentration of 0–40 µg/L (2 µg/L was a step), with a gradient that referred to the concentration range of *Rose Oxide* in real wine [16,17]. The sample was refrigerated at 4 °C after being prepared, sealed by the sealing film, and tested quickly and timely to reduce the volatilization of aroma substances.

De-aromatic wine solution: Ripened Cabernet Franc grapes were picked to make wine using 200 mg/L of Saccharomyces cerevisiae at 25–27 °C after hand-destemming and crushing. After the alcohol fermentation, the vacuum rotary evaporator was used to finish the deodorization procedure optimized based on Margaux Cameleyre [18]; the parameters of the rotary evaporator were set to 50 rpm and 30 °C, and two rounds of rotational evaporation were performed. For the first time, 500 mL of the original wine was spun in a water bath for 1.5 h. The matrix and ethanol water fractions of the wine sample were collected and blended to form 500 mL of initial de-aromatic wine, following which the second spinning was continued with identical parameters and was replenished with 12% ethanol–water to 500 mL again after 1.5 h. The matrix of de-aromatic wine and the distillate of anhydrous ethanol were collected and blended into the final de-aromatic wine. *Rose Oxide* was added to the de-aromatic wine, and the gradient settings and precautions were the same as those of the simulated wine samples.

Spectral data acquisition: The NIR spectra of different samples were recorded on a Fourier transform–near-infrared spectrometer equipped with an indium gallium arsenide detector from 11,500 to 4000 cm$^{-1}$. The temperature of the samples was equilibrated at 30 °C in the instrument. Each sample was scanned for 32 s with a spectral resolution of 8 cm$^{-1}$.

2.3.2. Data Pre-Processing and Outlier Rejection

The analysis process of spectral signals would be interfered with by the redundant information, spectral overlap, and baseline drift due to the complexity of the wine matrix. Min–max normalization (MMN) and vector normalization (VN) methods can effectively reduce redundant information and eliminate the effects of changes in the spectra, such as light range changes or sample dilution. First derivative (FD) and second derivative (SD) methods can eliminate the effects of baseline drift or smoothing background interference, distinguish overlapping peaks, and provide higher resolution and sharper spectral profile changes than the original spectra [19]. Thus, the above four methods were used to process the spectral signals of *Rose Oxide* in model wine and de-aromatic wine. For a spectral signal $x = (x_1, x_2, \ldots, x_n)$, the equations of MMN, VN, FD, and SD were as follows:

$$x_i^{\text{MMN}} = \frac{x_i - x_{\min}}{x_{\max} - x_{\min}} \tag{1}$$

$$x_i^{\text{VN}} = \frac{x_i}{\sqrt{\sum_{i=1}^{n} x_i^2}} \tag{2}$$

$$x_i^{\text{FD}} = \frac{x_{i+g} - x_i}{g} \tag{3}$$

$$x_i^{\text{SD}} = \frac{x_{i+g} - 2x_i + x_{i-g}}{g^2} \tag{4}$$

where $x_i$ was the $i$-th vector of $x$, $x_{\min}$ was the minimum vector of $x$, $x_{\max}$ was the maximum vector of $x$, and $g$ was the window width.

During the acquisition of *Rose Oxide* spectral signals, human and instrumental errors may cause some signals to deviate severely from the true value, resulting in outliers. In this study, principal component analysis (PCA) was used to shine the simples upon the low-dimensional space, then the outliers would be found according to the Hotelling $T^2$ statistic under the coordinate of first and second principal components [20]. The confidence level of $T^2$ detection was calculated as follows:

$$x^T(t)P\Lambda^{-1}P^T x(t) \leq \delta_{T^2} \tag{5}$$

$$\delta_{T^2} = \frac{(N-1)(N+1)}{N(N-K)}F \tag{6}$$

where $\delta_{T^2}$ was the confidence level of $T^2$ detection, $x(t)$ was the input matrix at the time $t$, $\Lambda$ was the covariance matrix, $N$ was the number of principal components, $k$ was the kth principal component, and $F$ was the F-distribution.

In this study, two $T^2$ confidence intervals with 99% and 95% were used to reject outliers. The specific screening criteria were as follows: samples located outside the 99% confidence interval were directly judged as outliers, and samples located between the two confidence intervals of 95% and 99% were judged as pending values and had to be validated to decide whether to reject them, while those located within the 95% confidence space were judged as excellent values and could be used for subsequent modeling.

### 2.3.3. Synergy Interval Partial Least Squares Regression (Si-PLSR)

The NIR spectrum contained abundant information regarding the molecular vibration absorption of hydrogen-containing groups, with most of them being redundant and unrelated. The removal of these extraneous spectral bands may significantly reduce the input variables and improve the accuracy of the prediction model. Therefore, synergy interval partial least square (Si-PLS) was used to screen the synergy intervals reflecting the *Rose Oxide* content [21,22]. Then, the rapid quantitative detection method of the *Rose Oxide* content in wine would be available based on the selected intervals. The specific calculation process was referred to [23]:

Step 1: Constructing PLSR models in the range of 11,500–4000 cm$^{-1}$ for *Rose Oxide*. The root means square error of cross validation ($RMSE_{CV}$) was calculated as:

$$RMSE = \sqrt{\frac{1}{n}\sum_{i=1}^{n}(y_i - y')^2} \tag{7}$$

where $y_i$ was the measured value of the *i*-th sample, $y'$ was the predicted value of the *i*-th sample, and $n$ was the number of samples.

Step 2: Dividing the spectral region of 11,600–4000 cm$^{-1}$ into 19 equal-width subintervals into steps of 400 cm$^{-1}$. Establishing the regression model of each subinterval by the PLS correction analysis.

Step 3: Selecting the subintervals, of which the $RMSE_{CV}$ were smaller than the $RMSE_{CV}$ calculated by step 1.

Step 4: Establishing the new Si-PLSR model based on the selected subintervals and evaluating the performance of the model.

### 2.3.4. External Validation

Twenty-one commercial wines were used to verify the accuracy of rapid detection models. First, headspace solid-phase microextraction combined with gas chromatography–mass spectrometry (HS/SPME-GC-MS) was used for quantifying the measured value of *Rose Oxide* [24]. The details were as follows:

SPME sample processing: Volatiles were extracted using solid-phase microextraction using DVB/CAR/PDMS fiber (50/30 μm film thickness, 2 cm Stableflex) assembled with a 57330-U holder (Supelco, Bellefonte, PA, USA). A wine sample (8 Ml), 2.0 g of NaCl, 2-octanol (final concentration was 400 μg/L), and a magnetic stirring bar were mixed in a 20 Ml glass vial. The vial was incubated in a thermostatic water bath to equilibrate for 15 min at 40 °C, then the fiber was exposed for 30 min at 40 °C. This was immediately followed by thermos-desorption of the extraction fiber in the GC injector for 5 min at 280 °C prior to GC-MS analysis. The extraction operation was repeated twice for each wine sample.

GC-MS analysis: GC-MS-QP2020 equipped with a DB-WAX capillary column (60 mm × 0.25 mm × 0.25 μm; Agilent J & W, Santa Clara, CA, USA) was used. The carrier gas was high-purity helium (99.999%) without shunt, and the gas flow rate was 1.5 Ml/min. The temperature of the GC capillary column was maintained as follows: 40 °C for 3 min, increase to 160 °C at a rate of 4 °C/min, followed by an increase to 220 °C at the rate of 7 °C/min, and this temperature was maintained for 10 min. We set the temperature of the inlet as 250 °C, the ion source as 220 °C, and the connecting rod as 200 °C. We set the energy of the electron impact source as 70 Ev. Electron ionization mass spectrometric data were acquired within the mass range of 35–350 *m/z* at 0.2 s intervals combined with the selected ion monitoring mode for the quantitative analysis.

Qualitative and quantitative analysis: A calibration curve for the pure standard was established to analyze the *Rose Oxide* content in the simulated wine solution by the above HS/SPME-GC-MS method. *Rose Oxide* was identified by comparing the retention times, retention indexes, aroma characteristics, and mass spectra with those of the standards available in the NIST 17.0 mass spectral library. The concentration of *Rose Oxide* was

quantitated by interpolating the relative area of the sample versus the area of the internal standard (2-octanol) using calibration curves previously established for pure standards. Then, the predicted value of *Rose Oxide* was obtained using the NIR model established in this study, and the validity and practical applicability of the model were judged by comparing the true values with the predicted values.

### *2.4. Data Analysis*

Microsoft excel was used for processing and preliminary analysis of spectral data, MATLAB R 2021b (MathWorks Inc., Natick, MA, USA) was adopted for spectrum preprocessing, Si-PLS analysis and model establishment, and Unscrambler X 10.4 (Camo Inc., Oslo, Norway) was used to eliminate outliers. Origin 2019 (OriginLab Inc., Northampton, MA, USA) was used for data drawing.

### 3. Results

### *3.1. Original Spectral Analysis of Model and De-Aromatic Wine*

The NIR spectral signals of the wine with added *Rose Oxide* are shown in Figure 1. The red curves expressed the model wine, and the blue curves expressed the de-aromatic wine. It could be observed that the different matrix backgrounds exerted a considerable effect on the height and width of the peaks in the curves, but they had little effect on the position of the peaks.

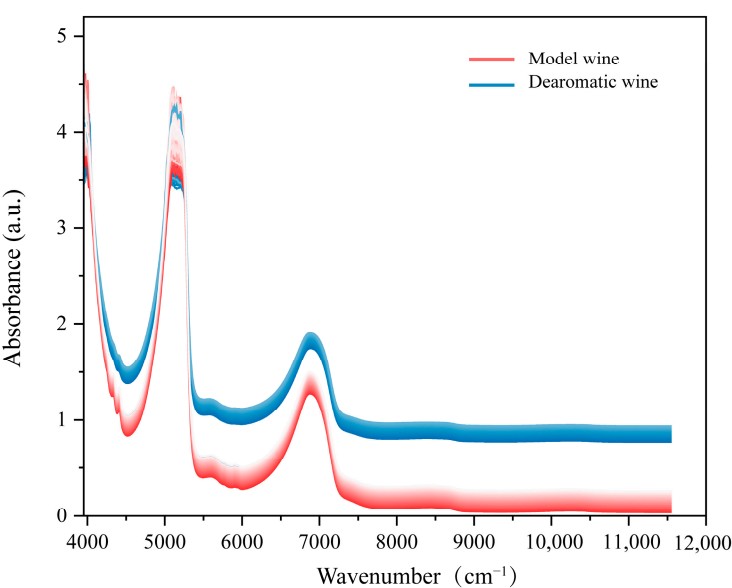

**Figure 1.** The raw spectra of different concentrations of *Rose Oxide* in simulated wine matrices and de-aromatic wine matrices.

The absorption peak of the de-aromatic wine was 0.30–1.00 (a.u.) higher than that of the model wine in 5500–11,600 $cm^{-1}$ and 0.00–0.50 (a.u.) higher in 4000–5000 $cm^{-1}$. However, their original spectra overlapped at 5000–5500 $cm^{-1}$, probably because of the characteristic absorption of ethanol in this interval, while the ethanol content of both matrices was consistent. Although the matrix solution was different, the change in the spectral curves caused by a different content of *Rose Oxide* could be observed at 4500–5500 $cm^{-1}$, 5500–6000 $cm^{-1}$, 6000–7500 $cm^{-1}$, and 8000–9000 $cm^{-1}$. Therefore, we tentatively speculated that the characteristic waveband of *Rose Oxide* is mainly located in the wave number range of these four regions.

### *3.2. Spectral Pre-Processing and Outlier Rejection*

The average data of all original spectral were pretreated using MMN, VN, FD, and SD. The spectral pre-processing results of *Rose Oxide* in model wine and de-aromatic wine are

shown in Figure 2. The spectral curves were narrower and smoother after pre-processing by the MMN and VN algorithm, but there was no significant difference between the two curves. The FD and SD algorithms separated the overlapping regions in the different spectral curves and intensified the absorption peaks around 4500 cm$^{-1}$, 5500 cm$^{-1}$, and 7500 cm$^{-1}$. The *RMSE* and the determination coefficients ($r^2$) estimated the availability of four pre-processing methods as shown in Table 1. Among them, MMN had the highest $r_{cv}^2$ value of 0.14 and the lowest $RMSE_{cv}$ value of 11.34 in de-aromatic wine, while VN showed the best performance, with an $r_{cv}^2$ value of 0.31 and $RMSE_{cv}$ value of 10.10 in model wine. Considering MMN exhibits a better performance for both spectral curves, and the de-aromatic wine was more complex than the model wine, MMN was selected as the best pre-processing method and applied in subsequent data analysis and model construction, while VN was only selected as the pre-processing method to explore the theoretical spectral features of *Rose Oxide*, which was not used for the subsequent establishment of the *Rose Oxide* prediction model.

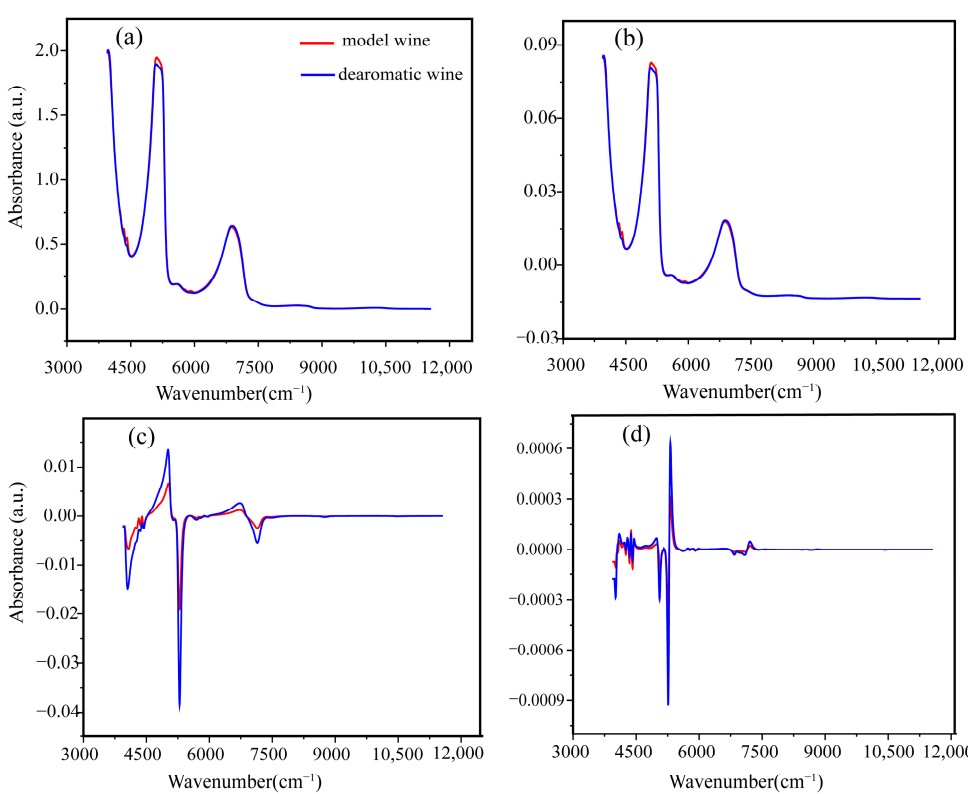

**Figure 2.** The spectral curves of *Rose Oxide* in model wine and de-aromatic wine based on (**a**) MMN; (**b**) VN; (**c**) FD; (**d**) SD.

**Table 1.** Results of spectral full-band modeling of model wine and de-aromatic wine by four pre-processing methods (before outlier rejection).

| Pre-Processing Methods | Model Wine | | | | De-Aromatic Wine | | | |
|---|---|---|---|---|---|---|---|---|
| | $r_c^2$ | $RMSE_C$ | $r_{cv}^2$ | $RMSE_{CV}$ | $r_c^2$ | $RMSE_C$ | $r_{cv}^2$ | $RMSE_{CV}$ |
| MMN | 0.75 | 6.37 | 0.23 | 10.70 | 0.04 | 12.10 | 0.14 | 11.34 |
| VN | 0.78 | 5.97 | 0.31 | 10.10 | 0.06 | 11.90 | 0.03 | 12.30 |
| FD | 0.51 | 8.85 | 0.12 | 11.40 | 0.05 | 12.00 | 0.05 | 12.00 |
| SD | 0.14 | 11.34 | 0.09 | 11.80 | 0.06 | 11.90 | 0.09 | 11.80 |

Note: $r_c^2$, the correlation coefficient of calibration set (the closer to 1, the better); $RMSE_C$, the calibration set root mean square error; $r_{cv}^2$, the correlation coefficient of the cross-validation set; $RMSE_{CV}$, root mean square error of calibration set (the smaller the better).

The outliers were found by the Hotelling $T^2$ statistics in the coordinate of PC1 and PC2, as shown in Figure 3. Out of a total of 133 model wine samples (No. 1 to 133), 3 samples (No. 90, 122, and 130) were excluded as outliers directly at a 99% confidence level. Sample No. 1 was distributed between the 95% and 99% confidence spaces and needed to be verified. As shown in Table 2, $r_p^2$ increased from 0.44 to 0.47, and $RMSE_P$ decreased from 9.03 to 8.67 after excluding the No. 1 sample, which indicated the it was an outlier. Out of a total of 133 de-aromatic wine samples (No. 1 to 133), 4 samples (No. 3, 63, 71, and 99) were outside the 99% confidence interval and were rejected directly. Additionally, five new samples (13, 31, 47, 109, and 128) were distributed between the 95% and 99% confidence spaces and needed to be verified. As shown in Table 2, the $r_p^2$ values increased, and the $RMSE_P$ values decreased by excluding samples No. 13, 109, and 128, while the model effect worsened after excluding samples No. 31 and 47. Therefore, samples No. 13, 109, and 128 were defined as outliers and samples No. 31 and 47 passed the verification. After removing the outliers, the *RMSE* values were reduced and the $r^2$ values were enhanced in most models (compared to the results in Table 1), suggesting that outlier rejection may improve the accuracy and stability of the prediction model. However, the model after pre-processing and outlier removal still did not work well because an *RPD* less than 1.5 meant a poor prediction performance. This may be caused by the interference of unrelated information in a full waveband. Thus, the feature wavebands of *Rose Oxide* in wines have to be screened out.

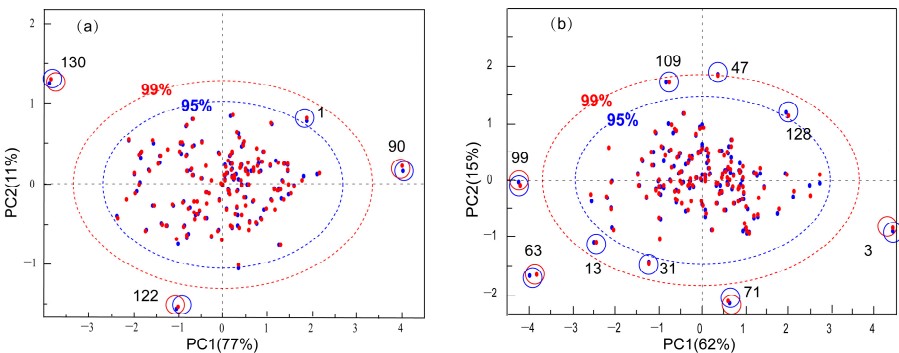

**Figure 3.** The outlier analysis by the Hotelling $T^2$ statistics in the coordinate of PCA. (**a**) The model wines; (**b**) the de-aromatic wines.

**Table 2.** Results of model validation after outlier rejection between 95% and 99% confidence space.

| | Sample Exclusion | $r_c^2$ | $RMSE_C$ | $r_p^2$ | $RMSE_P$ | $RPD$ |
|---|---|---|---|---|---|---|
| Model wine | All samples (except number 90, 122, and 130) | 0.78 | 5.93 | 0.44 | 9.03 | 1.33 |
| | 1 | 0.79 | 5.79 | 0.47 | 8.67 | 1.38 |
| De-aromatic wine | All samples (except number 3, 63, 71, and 99) | 0.61 | 7.95 | 0.11 | 11.60 | 1.06 |
| | 13 | 0.70 | 7.06 | 0.14 | 11.30 | 1.08 |
| | 31 | 0.67 | 7.32 | 0.09 | 11.70 | 1.05 |
| | 47 | 0.57 | 8.37 | 0.08 | 11.80 | 1.04 |
| | 109 | 0.63 | 7.82 | 0.18 | 11.10 | 1.10 |
| | 128 | 0.71 | 6.86 | 0.22 | 10.80 | 1.13 |

Note: $r_p^2$, the correlation coefficient of the validation set; $RMSE_P$, validation set root mean square error; *RPD*, the ratio of prediction to deviation (*RPD* < 1.5: poor model. 1.5 ≤ *RPD* < 2.5: general model. 2.5 ≤ *RPD* < 5: good model. *RPD* > 5: excellent model).

### 3.3. Si-PLS Analysis

*Rose Oxide*, as a monoterpene cyclic ether compound with a tetrahydropyran ring, is attached to a methyl and an isobutylene group, and possesses C=C, C–H, C–O–C, –CH$_3$, and –CH$_2$ functional groups. In this study, the Si-PLS method was used to extract the feature bands. The full wavenumber (11,600–4000 cm$^{-1}$) range was divided into 19 equal subintervals, with each subinterval modeled separately. The experimental results were shown in Figure 4. The $RMSE_{CV}$ values for subintervals 8, 11, 12, 13, 14, 15, 18, and

19 were smaller than the $RMSE_{CV}$ values (10.7 and 12.4) modeled for the full band in model and de-aromatic wine, while the remaining subintervals had lager $RMSE_{CV}$ values, which suggested that the model built with these eight subintervals would be better. The eight subintervals of two different background matrices were identical, indicating that they contained the feature information of *Rose Oxide*. Indeed, the references regarding the function group and spectral structure of *Rose Oxide* were limited. However, Davis et al. [25] conducted a study on the same functional groups in other chemicals, such as alkanes, alkenes, ethers, and tetrahydropyran, as shown in Table 3. It can be shown that the overtones of $CH_3$ stretching and deformation modes were largely responsible for the strong absorption region of 5901–5909 $cm^{-1}$ (the first overtone) and 8264–8696 $cm^{-1}$ (the second overtone), while the combination C–H stretching vibration bands of the $CH_3$ group were at 4100, 4395, 4400, 4500–4545, 5520, 5814, 7355, and 7263 $cm^{-1}$ in alkanes. The first overtone C–H stretching of the C–H=group bands were at 6100–6200 $cm^{-1}$, while the combination C=C stretching bands at 4482, near 4600, 4670–4780, and 6130 $cm^{-1}$, were found in alkenes. Moreover, the C–H stretching bands of CH and $CH_2$ functional groups in tetrahydropyran at 5565–6150 and 8040–9320 $cm^{-1}$ were assigned to the first and second overtones, respectively, while bands at 3885–4795, 6500, and 7500 $cm^{-1}$ were assigned to the combination regions. The C–O–C group was readily identified by the second overtone bands at 8300 and 8495 $cm^{-1}$ and the combination bands at 6400–7515 $cm^{-1}$ in ethers.

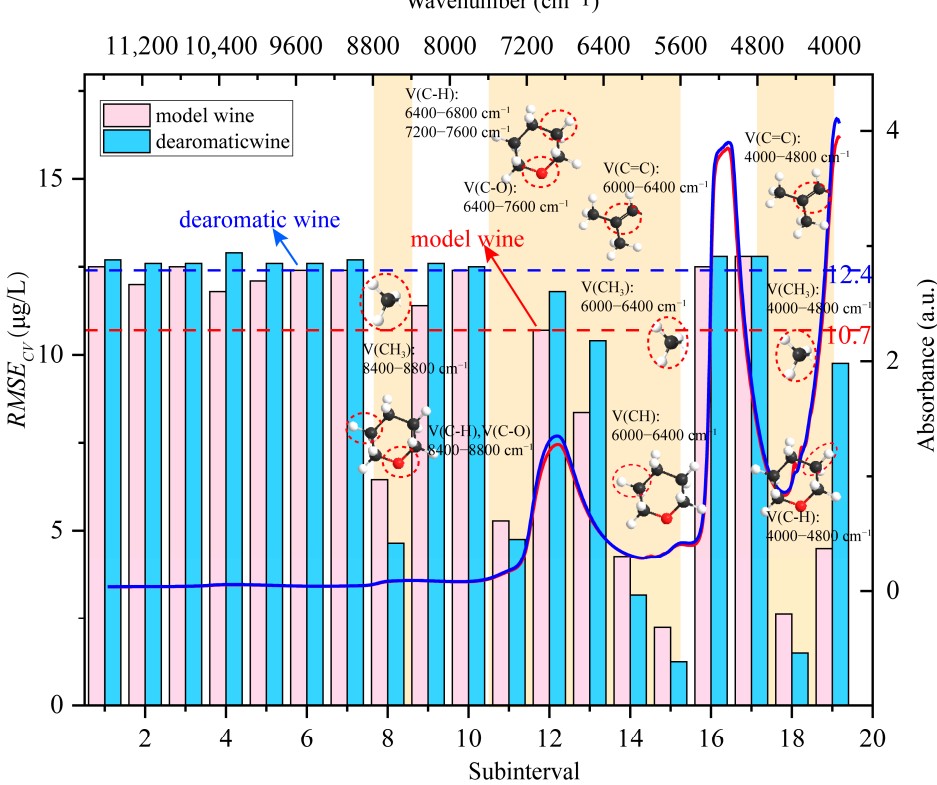

**Figure 4.** The $RMSE_{CV}$ values of Si-PLS based on each subinterval, and the feature wavebands analysis for the functional groups.

**Table 3.** Spectral mapping analysis of functional groups in similar substances of *Rose Oxide*.

| Chemicals | Assignment Groups | Wave Numbers (cm⁻¹) | | |
|---|---|---|---|---|
| | | First Overtone | Second Overtone | Combination Regions |
| Alkanes | V(C–H) | 5555–5882 | 8264–8696 | 6666–7090, 4545, and 4500 |
| | V(–CH₂–) | Near 6135 | Near 8290 | 4545 and 4525 |
| | V(–CH₃) | 5901–5909 | 8264–8696 | 4500–4545, 4395, 4100, 4400, 5520, 5814, 7355, and 7263 |
| Alkenes | V(C–H=) | 6100–6200 | | |
| | V(=CH₂) | | About 9260, 8787–9009, and 9091 | |
| | V(C=C) | | | 4482, near 4600, 4670–4780, and 6130 |
| Tetrahydropyran | V(C–H) | 5565–6150 | 8040–9320 | 3885–4795, 6500, and 7500 |
| | V(C–H) | | | 3800–4500 and 6400–7515 |
| Ethers | V(–CH₂–) | 5690 and 5790 | | |
| | V(–CH₃) | 5898 and 5910 | | |
| | V(CH–O–) | | 8300 | 6400–7515 |
| | V(CH₂–O–) | | 8495 | |

In this study, eight subintervals (4000–4400 cm⁻¹, 4400–4800 cm⁻¹, 5600–6000 cm⁻¹, 6000–6400 cm⁻¹, 6400–6800 cm⁻¹, 6800–7200 cm⁻¹, 7200–7600 cm⁻¹, 8400–8800 cm⁻¹) were recognized as the characteristic bands of *Rose Oxide* using the Si-PLS method. According to the assignments of the relevant groups mentioned in Table 3 and the eight characteristic intervals identified using the Si-PLS method, the group assignments of chemical structures in *Rose Oxide* were shown in Table 4. We observed that bands at the wave number region 5600–6000 cm⁻¹ were due to the first overtone of C–H stretching in the tetrahydropyran ring and methyl group, as well as the combination of the C–H stretching of the CH₃ function groups. Furthermore, bands between 6000 cm⁻¹ and 6400 cm⁻¹ were assigned to the first overtone C–H stretching of the C–H= group and the combination C=C stretching in isobutyl and the wavenumber of 8400–8800 cm⁻¹ belonged to the second overtone C–H stretching and C–O stretching in the tetrahydropyran ring, as well as the C–H stretching vibration in the methyl group. For the combination regions, 4000–4800 cm⁻¹, 6400–6800 cm⁻¹, and 7200–7600 cm⁻¹ were assigned to the C–H stretching vibration, while 6400–7600 cm⁻¹ was assigned to the C–O stretching vibration. These represented the spectral fingerprint information of *Rose Oxide* and are important for spectral identification and modeling applications.

**Table 4.** Group assignment of different chemical structures in *Rose Oxide*.

| Chemical Structure | Assignment Group | Wave Numbers (cm⁻¹) | | |
|---|---|---|---|---|
| | | First Overtone | Second Overtone | Combination Regions |
| Tetrahydropyran ring | V(C–H) | 5600–6000 | 8400–8800 | 4000–4800, 6400–6800, and 7200–7600 |
| | V(C–O) | | 8400–8800 | 6400–7600 |
| Methyl | V(CH₃) | 5600–6000 | 8400–8800 | 4000–4800, 5600–6000, 7200–7600 |
| Isobutyl | V(C–H=) | 6000–6400 | | |
| | V(C=C) | | | 4400–4800, 6000–6400 |

Nevertheless, the individual intervals included limited information and did not completely reflect the absorption properties of the *Rose Oxide* spectra. Therefore, subintervals 8, 11, 12, 13, 14, 15, 18, and 19 (corresponding to wave numbers: 8800–8400 cm⁻¹, 7600–5600 cm⁻¹, and 4800–4000 cm⁻¹) were selected as joint intervals and re-modeled using the PLSR method. According to the results of joint interval modeling shown in

Table [5](), the $r_c^2$ and $RMSE_c$ were 0.97 and 2.22 for the model wine and 0.97 and 2.36 for the de-aromatic wines, respectively; the $r_{cv}^2$ and $RMSE_{CV}$ were 0.96 and 2.55 for the model wine and 0.96 and 2.33 for the de-aromatic wines, which significantly improved the stability and predicted the accuracy of the model. The considerable improvement in the *RPD* value from 0.99 to 5.24 in de-aromatic wine indicated that a screened joint interval excluded a large amount of irrelevant information and condensed the spectral information of *Rose Oxide*, which laid the foundation for a further investigation of the spectral characteristics of its molecular groups and chemical bonds. The results of the best prediction model for the *Rose Oxide* were presented in Figure [5](). It was apparent that the validation data (cross-validation method used in this study) were in good agreement with the resulting model. The correlation between the values tested by HS/SPME-GC-MS and the NIR calibration for the different wine substrates was good, and the models showed a satisfactory fitting result and predictive ability.

**Table 5.** Results of full-band and joint interval modeling of model wine and de-aromatic wine.

| Interval Combinations | Model Wine | | | | | De-Aromatic Wine | | | | |
|---|---|---|---|---|---|---|---|---|---|---|
| | $r_c^2$ | $RMSE_C$ | $r_{cv}^2$ | $RMSE_{CV}$ | $RPD$ | $r_c^2$ | $RMSE_C$ | $r_{cv}^2$ | $RMSE_{CV}$ | $RPD$ |
| Full waveband | 0.75 | 6.37 | 0.23 | 10.70 | 1.19 | 0.04 | 12.10 | 0.14 | 11.34 | 0.99 |
| Joint interval | 0.97 | 2.22 | 0.96 | 2.55 | 4.78 | 0.97 | 2.36 | 0.96 | 2.33 | 5.24 |

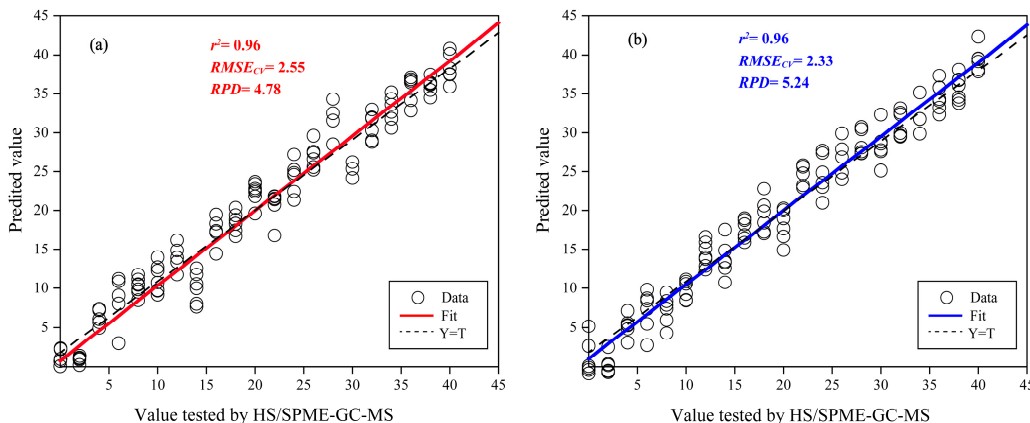

**Figure 5.** The results of the best prediction model for *Rose Oxide*. (**a**) The model wines; (**b**) the de-aromatic wines.

### 3.4. External Validation

The external validation aimed at estimating the predictive ability of the model based on a sample set that has not been included in the modeling process. In this study, the external validation of the PLS model for *Rose Oxide* in de-aromatic wine was conducted with the set of 21 samples, as shown in Table [6](). The $r_p^2$ was 0.84 (higher than 0.80), indicating that the PLS models for *Rose Oxide* based on NIR spectra explained 84.00% of the variation in the data. The *RPD* value obtained for the *Rose Oxide* in the external validation was 2.36 (higher than 1.50) and the $RMSE_P$ value was 2.72, indicating the good prediction capacity of the NIR models for *Rose Oxide* in real wines. The regression equations are also presented in Table [6](). It could be obtained that the *Rose Oxide* values tested by HS/SPME-GC-MS and the NIR calibration were similar. The results of external validation showed that this model could predict the *Rose Oxide* content of real wine to some extent, although the amount of information that can be explained was limited compared to the model built from the calibration set samples. On the one hand, the presence of other aromatic substances in real wine may affect the feature waveband of *Rose Oxide*, while on the other hand, it might be caused by the interference of other chemicals in the matrix. In conclusion, improving the

accuracy of the external validation of the model is important for the transfer and application of the model, which still requires extensive research in the future.

**Table 6.** External validation of the established PLS models based on de-aromatic wine for *Rose Oxide* (μg/L) in real wines.

| Spectral Number | External Validation | | | |
|---|---|---|---|---|
| | $RMSE_P$ | $RPD$ | $r_p^2$ | Regression Equation |
| 21 | 2.72 | 2.36 | 0.84 | $y = 0.717x + 3.5288$ |

## 4. Discussion

### 4.1. Spectral Band Allocation of Rose Oxide

The NIR technology is also called "black box" technology. Most studies pay little attention to the connection between the chemical groups and spectral wavebands of substances in "black box". In fact, it is still beneficial to the analysis and application of near-infrared spectroscopy to master the distribution of the organic compounds near-infrared band. In this study, the near-infrared waveband allocation of *Rose Oxide* was analyzed based on the model wine substrate, and eight feature waveband subintervals were screened using the Si-PLS method for associating with the chemical groups of *Rose Oxide*. In fact, it is difficult to accurately attribute the near-infrared band because the near-infrared band may be the combination of several different fundamental frequency double and harmonic spectrum bands, and there is no sharp peak and baseline separation of spectral peaks, mostly overlapping peaks and wide peaks. In this study, the simulation of wine substrate and the Si-PLS method were adopted to avoid the above defects. On one hand, the model wine is a simple matrix with alcohol and pH values consistent with real wine, which reduces the interference of other chemical components while simulating the actual situation as much as possible. On the other hand, although it is hard to accurately locate the near-infrared band, the distribution range can be expanded by screening the feature bands in the form of a joint interval so that the broad peaks of near-infrared can be basically distributed in the sub-interval. In the process of chemical group allocation, the distribution of chemical bonds will always be the focus because the absorption of organic matter in the near-infrared band is generally caused by various chemical bond stretching vibrations. In this study, the frequency doubling and co-frequency absorption of tetrahydropyran rings, C–H bonds, C–O bonds, and C=C bonds at different positions on the methyl and isobutene groups of rose ether were assigned to eight selected characteristic bands. It can be observed that wavebands 5600–6000 cm$^{-1}$ were related to the first overtone C–H stretching in the tetrahydropyran ring and methyl group, as well as the combination C–H stretching of the CH$_3$ function groups, which has been substantiated by the results of other studies. For example, Tosi and Pinto [26] found that bands near 5905 cm$^{-1}$ in all the hydrocarbons can be attributed to the methyl group. Burns and Ciurczak [27] showed that 5700 cm$^{-1}$, 5810 cm$^{-1}$, and 5900 cm$^{-1}$ were due to the 2v (C–H) vibration of the CH$_2$ functional group of cyclohexane and 2v (C–H) of the CH$_3$ groups of hydrocarbons with the methyl group. Bands between 6000 cm$^{-1}$ and 6400 cm$^{-1}$ were assigned to the first overtone C–H stretching of the C–H=group and the combination C=C stretching in isobutyl. As described by Gerasimov and Snavely [28], 6120, 6130, 6140, and 6200 cm$^{-1}$ corresponded to the CH stretching bands of vinyl (CH$_2$=CH–) and vinylidene (CH$_2$=C<), which was consistent with the results obtained in this study. In addition, the distribution of other characteristic bands has been confirmed by relevant studies. There are also studies that used the fundamental frequency of chemical substances in the mid-infrared region to calculate their frequency doubling and frequency co-absorption bands in the near-infrared region, which can provide a new direction for the near-infrared spectral band attribution analysis of *Rose Oxide*.

### 4.2. Potential of Near-Infrared Spectroscopy Models of Rose Oxide

De-aromatic wine is often used to explore the perceptual interaction among aromas in wine because it is a modeling background substrate obtained from real wine based on a strict de-aromatic procedure. Except for the absence of aroma substances, the non-volatile substrate is consistent with real wine [18]. In this study, the prediction model of *Rose Oxide* was established based on the de-aromatic wine matrix. The experiment shows that the prediction model of *Rose Oxide* can explain 84% of the information with an *RPD* value greater than 1.5, indicating that the method had a certain feasibility. Indeed, many studies have taken the near-infrared detection of volatile aroma substances into consideration. For example, NIR technology combined with the PLS method was used to detect esters and higher alcohols in wine, and it achieved a good prediction [9]. In addition, NIR technology was used to detect volatile aroma substances such as esters and short-chain fatty acids in Riesling wine, and the model established based on PLS has also shown good prediction results [29]. A NIR correction model of oak volatiles was established in dry red wine using the PLS method, of which the $r^2$ was greater than 0.86 and the *RPD* was greater than 1.5 [30]. These results indicate that NIR spectroscopy can be used for the rapid detection of volatile aroma substances in wine. However, the majority of previous research has been aimed at the near-infrared analysis of volatile substances with a relatively rich content in wine, such as esters and higher alcohols. There are many compounds in wine with a low content, such as *Rose Oxide*, but with a great aroma contribution. Their rapid detection is also particularly important. Under the condition of modeling based on real wine substrate, the aroma substances of *Rose Oxide* are often ignored because the lengthy pre-treatment will lead to the loss of *Rose Oxide* in the wine sample, which would result in an error in the test results. In this study, a modeling sample set was constructed based on the standard for *Rose Oxide* in different concentrations and de-aromatic wine substrates. Under the background of a substrate simulating real wine samples to the greatest extent, the measured values of *Rose Oxide* could be accurately obtained, and a wide range of modeling concentrations could be obtained, which had a good universality for the rapid detection of the aroma substances of *Rose Oxide*. It provides methodological support for the near-infrared rapid nondestructive detection of similar aroma substances in wine.

### 5. Conclusions

In this study, NIR spectroscopy was used to address the waveband allocation and quantitative prediction analysis of *Rose Oxide*. First, MMN and VN were used to pro-process the spectral signal of the de-aromatic wine and the model wine, and PAC was used to find the outliers. Then, Si-PLS was used to select the related spectral subintervals of the *Rose Oxide*, which improved the accuracy of the prediction model. Finally, the prediction model of the *Rose Oxide* content in de-aromatic wine was established and verified in real wine. The prediction model with a high $r^2$ and low *RMSE* was effective for detecting the content of *Rose Oxide* in wines under certain conditions, and it provided methodology support for the quantitative analysis of other terpenes in wine using this method. Since the wine samples in this study were collected from the producing area in the wine region of the Ningxia Helan Mountain's East Foothill, whether the *Rose Oxide* model is applicable to the analysis of other producing areas or imported wine samples needs to be validated and optimized.

**Author Contributions:** Conceptualization, X.B.; methodology, X.B. and Y.X.; formal analysis, X.X.; resources, Y.X.; data curation, X.C. and B.D.; writing—original draft preparation, Y.X. and X.X.; writing—review and editing, X.B. and Y.T.; project administration, X.B. and Y.T.; funding acquisition, X.B. and Y.T. All authors have read and agreed to the published version of the manuscript.

**Funding:** This work was supported by the Natural Science Foundation of China (32202213), the Natural Science Foundation of China (31972199), the Shaanxi Science Fund for Distinguished Young Scholars (2020JC-22), and the Nature Science Foundation of Shaanxi (2022JQ-222).

**Data Availability Statement:** Not applicable.

**Conflicts of Interest:** The authors declare no conflict of interest.

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
