# Peer review of "Analysis of Near-Infrared Spectral Properties and Quantitative Detection of Rose Oxide in Wine"

_agronomy, doi:10.3390/agronomy13041123_

Round 1

Reviewer 1 Report

This study proposed an information-based analysis method for Rose Oxide in wine, which need not labor-intensive and cost-consuming as the traditional experiment. This study also analyzed the spectral characteristics and response mechanism of Rose Oxide to support their method. The paper is clear and logical. It still needs to be revised in the following aspects:

1. Keywords fail to show the important technologies and methods in the study, such as “Band assignment” was meaningless information. The core algorithm should be included in the keywords.

2. The authors have not analyzed the latest research or application progress of spectral technology in wine. The literatures cited for analysis was published in early years.

3. The method is described briefly. Specially, four preprocessing algorithms are unclear and lack key steps and equations.

4. Table and Figure are not standardized. They need to be modified according to the requirements of the Agronomy template, including position, line, font, etc.

5. English writing of the paper is poor. Some sentences have errors, such as singular and plural of noun. The paper needs polishing by institution to correct fundamental linguistic errors.

Author Response

Dear reviewer:

Thanks for your comments to our manuscript. Detailed response to your comments were shown in attachment.

Kind regards.

Author Response

(The authors gave the same response as above.)
